# In Vitro Blood Clot Formation and Dissolution for Testing New Stroke-Treatment Devices

**DOI:** 10.3390/biomedicines10081870

**Published:** 2022-08-03

**Authors:** Kayla Wood, Sam E. Stephens, Feng Xu, Alshaimaa Hazaa, James C. Meek, Hanna K. Jensen, Morten O. Jensen, Ranil Wickramasinghe

**Affiliations:** 1Department of Biomedical Engineering, University of Arkansas, Fayetteville, AR 72701, USA; ksw015@uark.edu (K.W.); sxs094@uark.edu (S.E.S.); fengx21@vt.edu (F.X.); alshaimaa.atif@gmail.com (A.H.); 2Interventional Radiology Clinic, University of Arkansas for Medical Sciences, Little Rock, AR 72205, USA; jcmeek@uams.edu; 3Departments of Radiology and Surgery, University of Arkansas for Medical Sciences, Little Rock, AR 72205, USA; hkjensen@uams.edu; 4Ralph E Martin Department of Chemical Engineering, University of Arkansas, Fayetteville, AR 72701, USA

**Keywords:** stroke, plasmin, blood clots, dissolution, degradation

## Abstract

Strokes are among the leading causes of death worldwide. Ischemic stroke, due to plaque or other buildup blocking blood flow to the brain, is the most common type. Although ischemic stroke is treatable, current methods have severe shortcomings with high mortality rates. Clot retrieval devices, for example, can result in physically damaged vessels and death. This study aims to create blood clots that are representative of those found in vivo and demonstrate a new method of removing them. Static blood clots were formed using a 9:1 ratio of whole sheep blood and 2.45% calcium chloride solution. This mixture was heated in a water bath at 37 °C for approximately one hour until solidified. Following clot solidification, human plasmin was introduced by various methods, including soaking, injection, and membrane perfusion, and the resulting dissolution percentages were determined. Different clot types, representative of the wide range found physiologically, were also manufactured and their dissolution characteristics evaluated. A method to reproducibly create blood clots, characteristic of those found in vivo, is essential for the production of stroke retrieval devices that can efficiently and effectively remove clots from patients with low mortality rates and little/no damage to the surrounding vessels.

## 1. Introduction

Coagulation is an important bodily biological process that ensures that large amounts of blood are not lost following trauma and injury. However, coagulation can also block and disrupt blood flow as unwanted blood clots, which may lead to serious health deterioration or death. Blood clots can occur in various places throughout the body, and of particular concern are those that form in or migrate to places where they block blood flow to the brain. The brain requires a consistent and uninterrupted blood supply, which is vital for sustaining its function. If a blood vessel responsible for carrying oxygen and nutrients to the brain either ruptures or becomes blocked, ischemic stroke occurs. The affected part of the brain cannot obtain the oxygen and nutrients it needs, causing brain cells to begin to die. In the United States alone, about 795,000 people die from the disease each year [1]. Current treatments of ischemic stroke include administration of tissue plasminogen activator (tPA) and/or mechanical thrombectomy [2,3]. 

Development of advanced clot removal devices, based either upon retrieval, dissolution, or some other method, will require significant ex vivo product design and validation. Access to reproducible and representative model blood clots is essential. In this study, we build upon previously reported techniques to provide a method for reliably producing blood clots representative of those found in vivo [4]. The study was further designed to use the manufactured clots to determine the effectiveness of human plasmin as a clot-busting drug under simulated local delivery conditions.

## 2. Materials and Methods

Emboli analogs (EAs) were formed using whole sheep blood, stabilized with 3.8% (*w*/*v*) sodium citrate solution at a 1:10 anticoagulant–blood ratio (Lampire Biological Laboratories, Pipersville, PA, USA) [4,5]. Clots were formed from fresh blood refrigerated at 2 °C for less than 21 days to ensure consistent coagulation. To form the clots, the blood was mixed with 2.45% (*w*/*v*) calcium chloride solution (1720-4, Ricca Chemical Co., Arlington, TX, USA) at a 9:1 ratio [4]. The blood mixture, 2 mL per clot, was then placed into an open-ended, uncoated borosilicate glass tube (inner diameter: 11.4 mm; outer diameter: 13 mm) that was plugged at one end with a cork stopper and left open to atmospheric air at the other end. Medical-grade plastic tubing may also be used for clot formation. The tube was gently shaken to ensure thorough mixing and then put into a 37 °C water bath, open end up, for 1 h to coagulate. All clots were formed statically, with no agitation during coagulation. The solidified blood clots were removed from the glass tube by gently flushing with 10× phosphate-buffered saline (PBS). Following removal, smaller pieces of the clot, approximately 20 mm long, were cut using a scalpel. Fresh clots were used immediately, whereas aged clots, which simulate older, firmer clots, were stored for 24 h in plasma at 3 °C [6,7]. Physiological clots may have varying red blood cell (RBC) content (both clot to clot and within a single example), which respond differently to lysis [8,9]. Therefore, clots with varying amounts of RBC were additionally created by first centrifuging at 130× *g* for 30 min, permitting the plasma and RBCs to be separated by pipette. The separated plasma and RBCs, along with calcium chloride solution, were then recombined in the desired ratios [10]. 

To investigate the effectiveness of dissolution, lyophilized human plasmin (Athens Research and Technology, Athens, GA, USA) was used with various methods of delivery to the clot. To prepare the human plasmin, 342 µL of deionized water per milligram plasmin was added for reconstitution after an initial centrifugation (5414C, Eppendorf North America, Enfield, CT, USA) at 2000 rpm for 2 min. A buffer consisting of 20 mM NaPO_4_ at pH 7.4, 10 mg/mL d-Mannitol, and 10 mg/mL NaCl was used to further dilute from 44 U/mL to the desired concentrations. One unit (abbreviated as U) is defined as the amount of enzyme that will hydrolyze 1 µmole of tosyl-Gly-Pro-Lys-pNa per minute at 25 °C [11].

Plasmin dissolution was first evaluated by soaking the clot. Deionized water and 5.5 U/mL plasmin (50 µL each) were added into separate centrifuge tubes along with a small piece of fresh clot each so that the clots were completely submerged. Every 5 min, an additional 40 µL of deionized water and 5.5 U/mL was added to each tube for replenishment. The tubes were continually agitated by gentle manual swirling. Deionized water was used as a control to test if any of the clot dissolved due to mechanical disruption rather than due to the action of the plasmin alone. The blood clots were blotted dry with absorbent tissue to remove any excess fluid on the surface. The post-soak clot weight was then compared to the pre-soak weight to determine the degree of degradation. The percent degradation was calculated for the soaked clots as well as all following methods via the following equation: [(weight of clot before plasmin dissolution−weight of clot after dissolution) ÷ (weight of clot before dissolution)] × 100%.

Various methods of plasmin injection were additionally tested. Injection experiments were conducted on fresh EAs, comparing 50 μL injections every 5 min over a 30 min period with a single 200 μL injection. In order to determine how much degradation occurred due to mechanical damage from the needle, additional water injections were performed as a control into separate, fresh EAs. The percent degradation was calculated every 5 min.

In addition to soaking and injection, plasmin was also introduced via a porous membrane fiber that was passed through the clot, as illustrated in Figure 1 (see detailed description below). The membranes were created in-house from polyethersulfone (PES) and had inner and outer diameters of 0.6 and 1 mm, respectively, with an average pore diameter of 0.22 µm [12]. Scanning electron microscope images (Figure 1c,d) were obtained and used to determine pore diameter [13]. Micrographs were acquired with a FEI Nova Nanolab 200 Duo-Beam Workstation (Nanolab, Hillsboro, OR, USA), using a 15 kV acceleration voltage. A hypodermic needle (25Gx5/8”, EXELINT International Co., Redondo Beach, CA, USA) was placed inside the membrane. To secure the needle to the membrane, cyanoacrylate glue (Loctite, Dusseldorf, Germany) was placed where the needle and the membrane met and allowed to fully dry. Membranes were introduced through the clots with the help of a guide wire, formed from 24 American Wire Gauge (AWG) wire, approximately 0.52 mm wide. The wire was cut to approximately 9 cm in length and the membrane was cut to approximately 8 cm. The wire was inserted into the membrane through the open end until it reached the needle tip. The membrane/guide wire was then manually inserted through the center of the blood clot. Once the wire and approximately 1 cm of the membrane extended through the other end of the clot, the guide wire was pulled free so that only the membrane remained in the blood clot. 

A 1 mL syringe was filled with the prepared human plasmin and purged of air before being attached to the needle–membrane assembly via Luer connectors. The syringe plunger was manually depressed until a small drop (about 50 µL) of the plasmin could be seen at the other end of the membrane, ensuring that air had been purged from the inside of the membrane. The end of the membrane was then clamped with a binder clip to prevent any additional plasmin from coming out of the opening at the end of the membrane, rather than perfusing into/against the clot. The blood clot with the inserted diffusion membrane was placed onto a syringe pump (KD Scientific 78-0100, Holliston, MA, USA) for consistent delivery of the plasmin.

To investigate the role of increasing plasmin concentration upon dissolution, plasmin concentrations of 1.375 U/mL, 2.75 U/mL, and 5.5 U/mL were tested on different fresh EAs. A membrane-perfusion rate of 2.7 mL/min was used. Further, the differences in degradation between these plasmin concentrations were calculated in order to determine whether there was a point of diminishing returns. 

To ensure that the plasmin effectively dissolved a range of different blood clot types, dissolution rates of fresh EAs were compared to those in EAs that had been aged for 24 h. For both fresh and aged EAs, 50 μL of plasmin at a concentration of 3.2 U/mL was injected every 5 min over a 30 min time period, with degradation calculated every 5 min. Additionally, fresh blood clots with varying RBC compositions, ranging from 0 to 80%, were membrane-perfused with 3.2 U/mL plasmin. Plasmin was perfused into these clots at 2.7 mL/min through membrane insertion, as described above, and observed over a period of three hours. Percent degradation was calculated at 30 min intervals over a 3 h period.

Statistical significance was determined by two-sided, unpaired Student’s *t*-tests. The null hypothesis is that there is no significant difference between clot degradations, whereas the alternative hypothesis is that a significant difference is present.

## 3. Results

Degradation percentages for the soaked clot fragments are shown in Figure 2. The deionized water-soaked clot showed a much smaller degradation than that of the plasmin-soaked clot, indicating that while some mechanical breakdown occurred (while keeping both clots moist), it was not the primary mechanism of clot degradation.

For injection-delivered plasmin, minor degradation was observed with the deionized water control (5–10%) (*n* = 6), indicating that while mechanical breakup does have an effect, it is small compared to the action of plasmin (p = 1.2 × 10^−3^). The degradations observed for multiple small injections and a single large injection were similar in magnitude and trajectory (*p* = 0.694).

Differences in degradation percentages due to membrane-perfused plasmin between various concentrations, taken at the same time points over a 30 min time course, are shown in Figure 3. Within the 1.375–2.75 U/mL range, degradation appears highly dependent upon concentration. Beyond this, diminishing returns are observed, where increasing concentration does not significantly increase degradation rate.

Plasmin injection degradation percentages in the fresh EAs was consistently higher than for the aged EAs by approximately 14 ± 3% (*n* = 6, *p* = 0.19). The degradation of clots with varying RBC content, by membrane-perfused plasmin, are shown in Figure 4. Degradations are highest for 0% RBC clots, with degradation decreasing as RBC content increases. Statistically significant (*p* < 0.05) differences were observed between all RBC-content clots except between 0% and 5%.

## 4. Discussion

Current treatment methods for stroke rely primarily upon either systemic treatment resulting in a chemical dissolution, or purely mechanical retrieval of the clot. Dissolution treatment utilizes the enzyme tissue plasminogen activator (tPA) to break up the thrombus [14]. Systemically delivered tPA converts plasminogen into plasmin, which is the main enzyme responsible for dissolving fibrin, one of the main building blocks of clots. Mechanical retrieval, on the other hand, utilizes catheter-based devices to remove the thrombus. Stent retriever devices, such as the Trevo^TM^ and Solitaire^TM^, are used to “grab” the clot by mechanical interference, allowing the clot to be pulled out along with the catheter [15,16]. Aspiration guide catheters, such as the Penumbra, provide suction to assist in removal [2,17,18]. Often an aspiration guide is used in combination with a stent retriever.

Despite having several accepted options, difficulties remain for stroke treatment. Dissolution via systemic delivery of tPA is very time-sensitive, with substantially improved outcomes, including treatment times shortened by as much as 15 min [19]. Administration of tPA beyond 4.5 h of stroke onset may even prove harmful as it increases risk of hemorrhagic transformation [20]. Unfortunately, studies have found that only about 15%–32% of ischemic stroke patients receive tPA in this time frame and, of these patients, only about 40%–50% are even eligible to receive tPA clinically [21]. Additionally, tPA is administered systemically, limiting the maximum dosage due to bleeding concerns [22,23]. Mechanical devices, in addition to being more invasive, face the constant threat of clot fragmentation, which may lead to distal re-embolization due to incomplete capture/retrieval [24]. Both chemical and mechanical treatment methods are further confounded by the fact that clots can vary widely in their composition. Categorized generally as fibrin-rich or platelet-rich depending upon their genesis and makeup, the responsiveness of these types to chemical dissolution varies, as does their structural integrity and thus susceptibility to mechanical retrieval methods [25,26,27,28,29].

Access to EAs in vitro dramatically simplifies the development of stroke treatment devices by significantly reducing the need for complex and expensive animal trials [4]. The associated reduction in logistical difficulty can lead to more rapid design iterations and decreased time-to-market and cost. Visualization of the treatment methods, normally achieved through fluoroscopy, can also be dramatically improved through the use of transparent phantoms for optical access [30].

The plasmin studies presented here showed efficient clot dissolution in a controlled environment. Controls using deionized water showed relatively small degradations, indicating that while mechanical breakup accounts for a small portion of the overall degradation, the vast majority is due to the action of the plasmin. Fresh EA degradation was approximately 30% after 5 min and 80% after 30 min. Anecdotally, it was observed that a single large dose of plasmin was more effective than steady, smaller doses in shorter time frames of 5 min or less, whereas the two were nearly equal over the longer term of above 20 min. Higher plasmin concentrations were more effective up to a point, with the dissolution rate stabilizing upwards of 3 U/mL. As the wetted area remains nearly constant, it may be that this area becomes saturated by protein past this point. Increasing the wetted area, perhaps by the addition of multiple membranes, may further increase the degradation rate. Unsurprisingly, clots with lower RBC content dissolved more readily. Aged clots, on the other hand, proved more challenging, having degradations 10%–15% lower than those for fresh EAs. These clots may require more agitation as part of the dissolution process. 

Though not FDA-approved for the treatment of ischemic stroke, plasmin has been evaluated in a number of human and animal studies. Plasmin dosage in in vivo work varies, but typically is in the 0.03–0.8 mg/min range [31,32]. The above-detailed plasmin injections of 50 and 200 µL correspond to approximately 0.018 and 0.073 mg of protein, respectively. Membrane-perfused plasmin, delivered at a concentration of 5.5 U/mL and a rate of 2.7 mL/min, translates to approximately 0.976 mg of protein per minute. While previously reported plasmin dosages are comparable to those used here, direct comparison is not possible due to a variety of factors, such as the lack of plasmin dilution by blood in our model, and differences in overall clot size. Despite this, the fact that significant clot degradations were observed with our modest doses bodes well for the use of plasmin as a thrombolytic agent.

## 5. Conclusions

The development and evaluation of stroke treatment methods, including mechanical retrieval as well as chemical dissolution, require reliable benchtop models. We have built upon our group’s previously published work to develop a repeatable method for creating EAs with a wide range of properties, including varying RBC content and clot age. These clots responded well to plasmin dissolution with several delivery mechanisms, including membrane perfusion through a permeable fiber. These results demonstrate the utility of such EAs towards the development and testing of improved stroke-treatment devices.

## Figures and Tables

**Figure 1 biomedicines-10-01870-f001:**
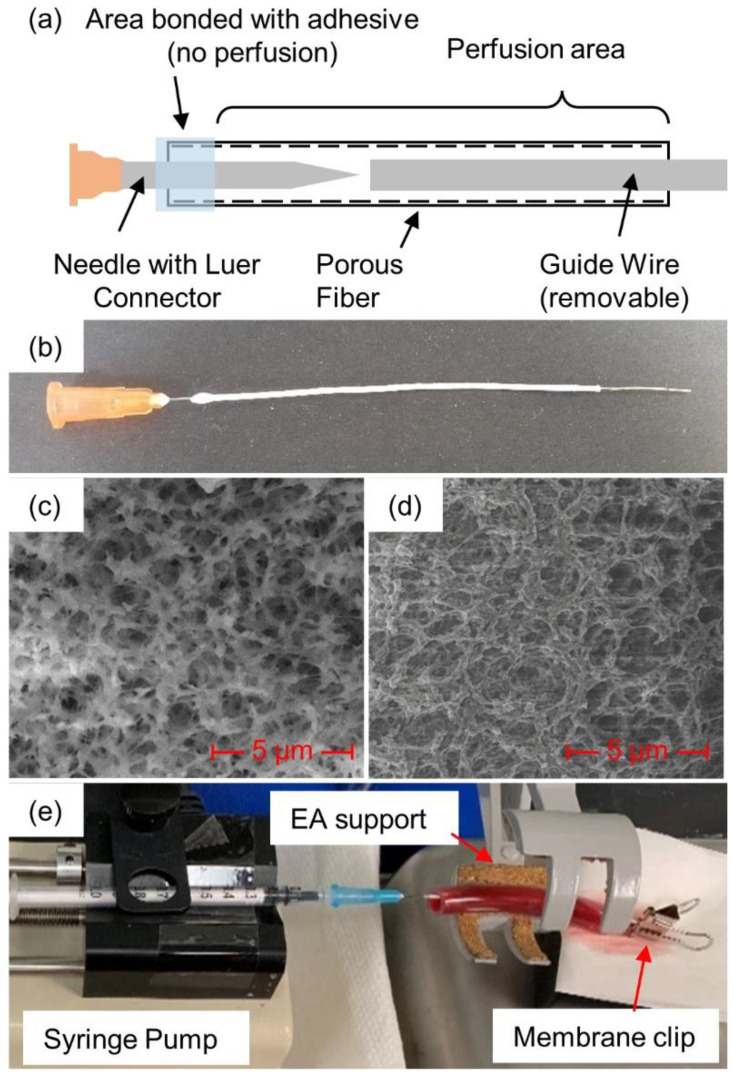
Plasmin-membrane-perfusion setup: (**a**) schematic view and (**b**) actual view of membrane, (**c**) SEM photos at 10,000× magnification of fiber cross-section and (**d**) inside surface, along with (**e**) full membrane-perfusion system with blood clot inside tube.

**Figure 2 biomedicines-10-01870-f002:**
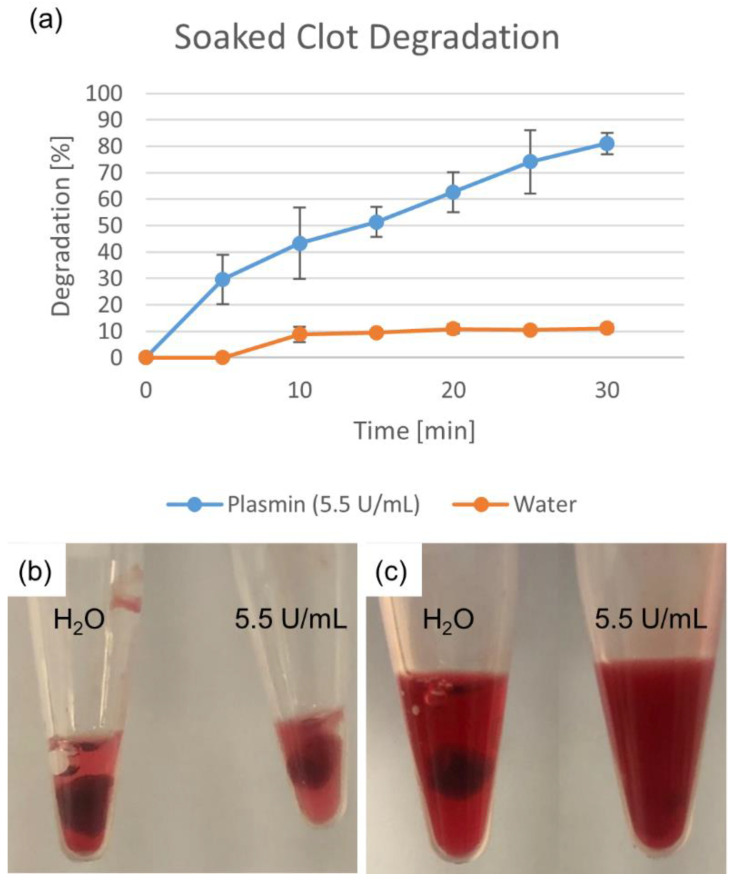
Degradation due to plasmin soaking. Deionized water degradation control accounts for mechanical breakdown. Percentage shown as mean ± standard deviation (*n* = 3). (**a**) Water- and plasmin-soaked clots are shown (**b**) initially and (**c**) after 25 min.

**Figure 3 biomedicines-10-01870-f003:**
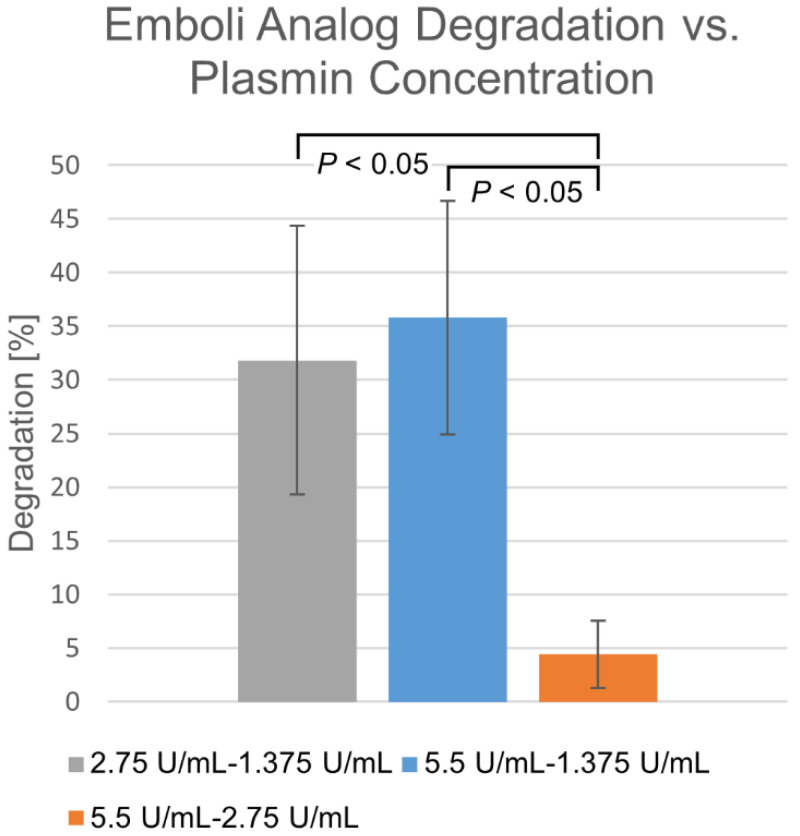
Differences in membrane-perfusion-delivered plasmin degradation at varying concentrations. Percentage shown as mean ± standard deviation (*n* = 6).

**Figure 4 biomedicines-10-01870-f004:**
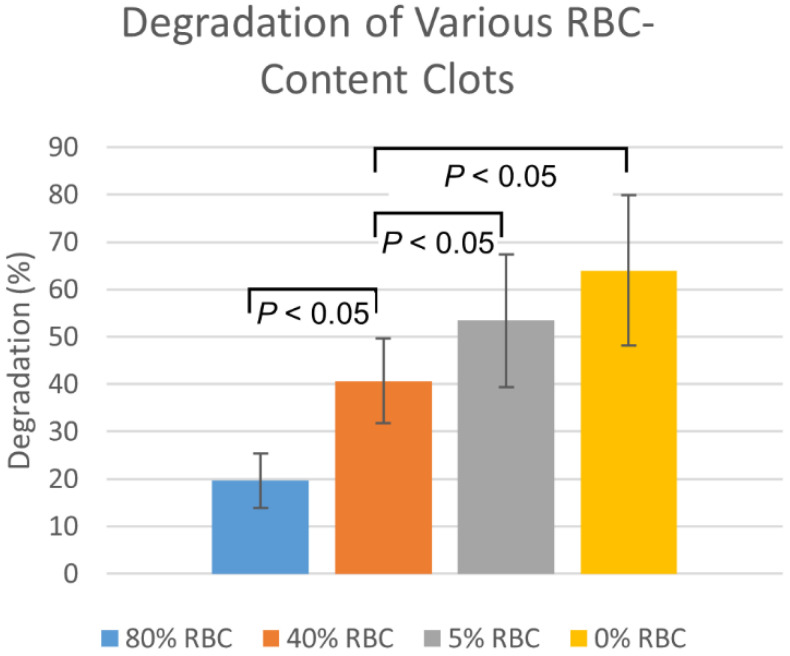
Membrane-perfused plasmin degradation of clots formed with varying red blood cell compositions. Values are reported as mean ± standard deviation (*n* = 12).

## Data Availability

Data are available from authors upon request.

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
