# Peer review of "In Vitro Blood Clot Formation and Dissolution for Testing New Stroke-Treatment Devices"

_biomedicines, 2022, doi:10.3390/biomedicines10081870_

Round 1

Reviewer 1 Report

The authors conducted an experiment to develop an in vitro blood clot model mimicking the in vivo blood clot formation. They showed that, in the model, blood clots not only formed but also dissolved upon exposure to human plasmin, introduced by soaking, injection, or membrane perfusion, in both a time-dependent and concentration-dependent manner.  

There are some comments.

Comments:

1.      Materials and Methods (Line 139 on page 4): “Statistical significance was determined by one-sided, unpaired Student’s t-tests.” However, based on the null and alternative hypotheses descriptions, a two-sided t-test would be needed. I would suggest restating the alternative hypothesis (for instance, higher degradations compared to control) or using a two-sided t-test.

2.      Results (Figure 3): This figure shows the degradations with plasmin concentrations between 1.375 and 2.75 U/mL, 1.375 and 5.5 I/mL, and 2.75 and 5.5 U/mL. However, Materials and Methods (Line 128 on page 4) stated, “plasmin concentrations of 1.375 U/mL, 2.75 U/mL, and 5.5 U/mL were tested.” I would suggest presenting the degradation at these concentrations in Figure 3.

3.      Results (Figure 4): I would suggest testing the differences in degradation between clots with 0 RBC and clots with 5% (or 40%) RBC.

4.      Discussion: In Materials and Methods (Line 128 on page 4), “plasmin concentrations of 1.375 U/mL, 2.75 U/mL, and 5.5 U/mL were tested--.” Has plasmin at these concentrations been effective in in vivo models? If yes, this would be another piece of evidence supporting the usefulness of this in vitro model.

5.      Discussion: Using the developed in vitro model, the authors showed that further increasing plasmin concentration beyond 2.75 U/mL failed to increase further the degradation rate (Line 160 on page 5). I would encourage the authors to discuss this finding more. For instance, was this phenomenon also observed in in vivo models (or even humans/patients)? 

Reviewer 2 Report

Dear authors,

Please reconsider the authors instructions guidelines for your article.

There are a lot of incongruences...

For example the references   and authors contributions are not ok as the style impose by the journal guidelines!

This reference is to old to be cited in this topic: Rymner, M.M., et al., Management of acute ischemic stroke: time is brain. Missouri medicine, 275 2010. 107(5): p. 333-337.

Simons, N., et al., Thrombus composition in acute ischemic stroke: a histopathological study 290 of thrombus extracted by endovascular retrieval. (0150-9861 (Print)). ??

Laughlin, M.E., et al., Development of Custom Wall-Less Cardiovascular Flow Phantoms with 297 Tissue-Mimicking Gel. Cardiovascular Engineering and Technology, 2021. ??

Round 2

Reviewer 2 Report

Dear authors,

your manuscript is enhanced and clear now.

the English style was improved.

you have clarified the actual terminology used.